# Adherence to Vaccines in Adult Patients with Immune-Mediated Inflammatory Diseases: A Two-Year Prospective Portuguese Cohort Study

**DOI:** 10.3390/vaccines11030703

**Published:** 2023-03-20

**Authors:** Candida Abreu, Antonio Martins, Fernando Silva, Gabriela Canelas, Lucia Ribeiro, Stefano Pinto, Antonio Sarmento, Fernando Magro

**Affiliations:** 1Department of Infectious Diseases, São João Hospital Center, Alameda Prof. Hernâni Monteiro, 4200-319 Porto, Portugal; 2Department of Medicine, Faculty of Medicine, University of Porto, Alameda Prof. Hernâni Monteiro, 4200-319 Porto, Portugal; 3Instituto de Inovação e Investigação em Saúde (I3S), Rua Alfredo Allen 208, 4200-135 Porto, Portugal; 4Instituto Nacional de Engenharia Biomédica (INEB), Rua do Campo Alegre 823, 4150-177 Porto, Portugal; 5Unit of Pharmacology and Therapeutics, Department of Biomedicine, Faculty of Medicine, University of Porto, Alameda Prof. Hernâni Monteiro, 4200-319 Porto, Portugal; 6Department of Gastroenterology, São João Hospital Center, Alameda Prof. Hernâni Monteiro, 4200-319 Porto, Portugal; 7Clinical Pharmacology Unit, São João Hospital Center, Alameda Prof. Hernâni Monteiro, 4200-319 Porto, Portugal

**Keywords:** vaccine adherence, vaccine prescription, inflammatory bowel diseases, rheumatological diseases, immunosuppressed patients

## Abstract

Background: Patients with immune-mediated inflammatory diseases (IMIDs) treated with immunomodulatory therapy present an increased susceptibility to infections. Vaccination is a crucial element in the management of IMID patients; however, rates remain suboptimal. This study intended to clarify the adherence to prescribed vaccines. Materials and methods: This prospective cohort study included 262 consecutive adults with inflammatory bowel disease and rheumatological diseases who underwent an infectious diseases evaluation before initiating or switching immunosuppressive/biological therapy. Vaccine prescription and adherence were assessed during an infectious diseases (ID) consultation using a real-world multidisciplinary clinical project. Results: At baseline, less than 5% had all their vaccines up-to-date. More than 650 vaccines were prescribed to 250 (95.4%) patients. The most prescribed were pneumococcal and influenza vaccines, followed by hepatitis A and B vaccines. Adherence to each of the vaccines ranged from 69.1–87.3%. Complete adherence to vaccines occurred in 151 (60.4%) patients, while 190 (76%) got at least two-thirds of them. Twenty patients (8%) did not adhere to any of the vaccines. No significant differences were found in the adherence rates of patients with different sociodemographic and health-related determinants. Conclusions: ID physicians can play a role in the process of increasing vaccine prescription and adherence. However, more data on patients’ beliefs and vaccine hesitancy, along with mobilization of all health care professionals and adequate local interventions, shall be considered to improve vaccine adherence.

## 1. Introduction

Over the last years, the treatment of patients with immune-mediated inflammatory diseases (IMIDs), such as inflammatory bowel disease (IBD) and rheumatological diseases, with biologic and other immunomodulatory drugs has resulted in increased disease management effectiveness. However, due to their mechanism of action, these drugs may increase the risk of infection, including severe and/or opportunistic infections. As such, vaccination is a key factor in protecting these patients from infectious diseases (ID) and improving their overall health status [1,2].

In general, vaccines are considered to be effective and safe. However, some particularities of the patients with IMIDs (treated with immunomodulatory, biologic, and small molecule drugs or their combinations) may pose special concerns regarding vaccine efficacy and contraindications of live attenuated vaccines. Several guidelines concerning vaccination in these patients are now available with the objective of improving the vaccination process in this specific context [2,3,4,5]. Even though adult patients with IBD and rheumatological diseases still have suboptimal vaccination rates [6,7]. Misconceptions about effectiveness and safety among health care providers and patients may explain, in part, the reported lower levels of vaccination in the setting of IMIDs [8]. From our perspective, further interventions, directed at both IMID patients and healthcare providers, are needed to raise awareness on this topic and clarify aspects related to the effectiveness and safety of vaccines.

In the context of IMID management, gastroenterologists and rheumatologists may not prioritize the discussion and prescription of vaccines or the screening of infectious diseases before prescribing immunomodulatory drugs. In this context, a multidisciplinary approach directed at patients with IMIDs, including infectious diseases (ID) and general and family physicians and focused on infection screening and prevention, may be beneficial for the overall management of these patients.

This prospective cohort study aimed to go beyond vaccine prescription and evaluate the adherence to vaccines in adult patients with IMIDs assessed in the context of a consultation with an infectious disease (ID) physician. This study was conducted using a real-world multidisciplinary clinical project established between gastroenterologists, rheumatologists, and ID physicians that was conceived to screen for and prevent infections in adult patients with IBD and rheumatological diseases.

Our manuscript aims to share our experience with this cohort and ultimately increase vaccination rates. We believe that a multidisciplinary approach directed towards patients with IMIDs is a robust strategy for surveillance and treatment. Infection screening and prevention are naturally the primary focus of ID physicians, making them well-suited for vaccine-related tasks. However, we went beyond merely prescribing vaccines and emphasized adherence to vaccine prescriptions in adult patients with IMIDs assessed in the context of a consultation with an infectious disease (ID) physician.

## 2. Materials and Methods

### 2.1. Settings

A prospective cohort study was conducted between January 2019 and December 2020 by enrolling adult patients (≥18 years of age) with IMIDs. Patients were referred to the ID Clinic from the Gastroenterology or Rheumatology departments. The referral was performed before starting immunosuppressive or biological therapy or, in the case of switching therapies. We enrolled all consecutively eligible patients who met the inclusion criteria and provided their informed consent to participate in the study during the study period.

In the ID consultation (IDC), patients were approached in terms of infectious disease screening, verification of the vaccination plan, vaccine prescription, and preparation for biological treatment. General and family physicians were also involved in the vaccination process, namely through vaccine administration and adherence monitoring, according to their availability. The aim of this study was to evaluate vaccine prescription in the setting of IDC as well as the adherence of adult patients with IMIDs to the prescribed vaccines.

### 2.2. Vaccine Prescription

Vaccine prescriptions were carried out in accordance with the recommendations of EULAR, AGA, and ECCO [2,3,4], as follows: Hepatitis A vaccine—patients with no previous disease or vaccination and no IgG antibody to hepatitis A virus; hepatitis B vaccine—patients with no previous disease or vaccination and no hepatitis B surface antigen, and patients with previous vaccination and with hepatitis B surface antibodies below 10 U/L; pneumococcal protein conjugate 13-valent vaccine (PCV13)—patients with no previous vaccine; pneumococcal polysaccharide 23-valent vaccine (PPV23)—patients with no previous vaccine and at least 2 months after PCV13; influenza vaccine—all patients during the flu season (once); tetanus, diphtheria and pertussis, and poliomyelitis vaccines—according to the Portuguese National Vaccination Plan; measles, mumps and rubella (MMR), and chickenpox vaccines—patients with no previous disease or vaccination and without IgG antibody (as long as no immunosuppression); zoster (live) and yellow fever vaccines—in specific relevant situations (as long as no immunosuppression); other vaccines from the Portuguese National Vaccination Plan—if not up-to-date and no contraindication was elicited.

The vaccines that were administered have been recorded in the national vaccine registry platform of the Portuguese National Health Care System. In rare cases, some vaccines may have been recorded in an individual vaccination book from the same system.

### 2.3. Data Collection

Demographic and clinical data were obtained through electronic medical records (EMR). The collected data included age, gender, education level, data on the IMIDs, ongoing immunomodulatory therapies, and vaccination history. Immunosuppressive drugs included azathioprine, 6-mercaptopurine, methotrexate, leflunomide, and steroids; biological drugs included anti-TNF, vedolizumab, and ustekinumab, as well as small molecules such as tofacitinib. Vaccine prescriptions in the IDC and adherence to prescriptions were registered. The outcome of interest was vaccine adherence, defined as complete (all vaccines), partial (at least one but not all), or null (none). Adherence to the hepatitis B vaccine was considered only if all prescribed immunizations were taken.

### 2.4. Statistical Analysis

SPPS software version 27 was used. Descriptive statistics were used to summarize the data, including frequencies and/or percentages for categorical variables and median and inter-quartile range (IQR) or mean and standard deviation (SD) for numerical variables. The Chi-square test and Fisher’s exact test were used for categorical variables, as appropriate. The nonparametric Mann-Whitney test or the independent-sample T-test was used for numerical variables, considering normality assumptions. A *p*-value < 0.05 was considered significant.

Quality appraisal using critical appraisal skill program (CASP) checklist for cohort studies was included (Appendix A).

### 2.5. Ethics Statement

The study was approved by the Ethics Committee of CHUSJ (356/2018). Informed written consent, in accordance with the Declaration of Helsinki, was obtained after explaining the procedures to each participant. The participants were able to leave the research at any time without any consequences, and the individuals who decided not to participate received the same treatment offered to participants. The method of data collection allowed for the integration of demographic information into the evaluation while ensuring confidentiality.

## 3. Results

Two hundred and sixty-two patients were included in the study. Sociodemographic and clinical data are summarized in Table 1. The most common IMIDs were Crohn’s disease (163 patients, 62.2%), ulcerative colitis (52 patients, 19.8%), and rheumatoid arthritis (21 patients, 8%). One-hundred and sixty-seven (63.7%) patients were already on immunomodulatory therapy on the first IDC; 46 (17.6%) were taking two drugs, and 11 (4.2%) were on three different drugs. Among these patients, 126 (75.5%) were on immunosuppressive drugs, 23 (13.8%) were on biologic drugs, and 18 (10.8%) were on both immunosuppressive and biologic drugs. The median time from immunomodulatory therapy initiation or switch to IDC was 9 (Q1 = 1; Q3 = 75) months.

Vaccine prescription and adherence are described in Table 2. Vaccines were considered up-to-date in 12 (4.6%) patients; 10 (83.3%) of these patients were already on immunomodulatory therapy. Of the 250 (95.4%) patients without up-to-date vaccines, 157 (63%) were already on immunomodulatory therapy. Among patients under immunomodulatory therapy (*n* = 167), 157 (94%) received vaccine prescriptions. Among the non-medicated patients (*n* = 95), 93 (98%) had vaccines prescribed.

Vaccines for the prevention of respiratory diseases [pneumococcal vaccines (PCV13 and PPV23) and influenza vaccine] were the most prescribed (Table 2). Adherence to these vaccines ranged from 69.1% for PPV23 to 87.3% for PCV13. Adherence to PCV13 was higher for those on immunosuppressive drugs compared to patients treated with biological drugs, with or without immunosuppressants (*p* < 0.001) (Table 3).

The hepatitis A vaccine was prescribed in 98 (39.2%) and the hepatitis B vaccine in 91 (36.4%) of the 250 patients. Considering hepatitis B, 37 patients had vaccine boosters, and 54 received the full series. Adherence to the hepatitis B vaccine was considered if all prescribed doses were taken. In this context, adherence to hepatitis A and B vaccines was 78.6% and 80.2%, respectively. Adherence was higher in patients who were prescribed hepatitis B vaccine boosters (75.7%) but also in those who were prescribed the full series (83.3%). Adherence to these vaccines was not statistically different between different therapy groups (Table 3).

Sixteen live attenuated vaccines (8 MMR, 4 zoster-live, 3 chickenpox, and 1 yellow fever) were prescribed to 14 patients, with two patients receiving two prescriptions each. Moreover, two patients on immunosuppressive drugs were prescribed live vaccines; one patient was on prednisolone 5 mg daily and the other suspended prednisolone to be vaccinated (Table 3). Even though 31 patients would be eligible (50 years old or older and not on immunomodulatory or immunosuppressive therapy), the zoster (live) vaccine was prescribed only to 4 patients due to financial constraints.

Overall, 151 (60.4%) patients had all the prescribed vaccines, and 39 (15.6%) had at least 2/3 of the vaccines prescribed. Twenty (8%) patients did not adhere to any of the prescribed vaccines, including nine patients who were only prescribed one vaccine. Patients that had complete adherence (*n* = 151, 60.4%) were compared to those that had partial or no adherence (*n* = 102, 40.8%) (Table 4). We could not find differences in vaccine adherence according to age, gender, education level, IMID group, time from diagnosis to IDC, therapy group, and time from therapy to IDC consultation (Table 4); also, no differences in vaccine adherence rates were found according to the same conditions in patients with complete vaccine adherence and in those with only partial adherence (data not shown).

## 4. Discussion

In this prospective study, we monitored vaccine prescription and adherence in IMID patients in the setting of an IDC performed before or during immunosuppressive or biological therapy. Overall, most of the patients (60.4%) had all the prescribed vaccines, and about 15.6% had at least 2/3 of the vaccines prescribed. Adherence to PCV13 was significantly higher in patients treated with immunosuppressive drugs.

### 4.1. Vaccine Prescription

In our study, due to a delayed referral in most of the patients, inactivated vaccines were prescribed after the most appropriate timing for administration, which would be before starting the treatment with immunosuppressants. This may have reduced the effectiveness of these vaccines and hindered the administration of live-attenuated vaccines such as MMR, chickenpox, and zoster (live), which are contraindicated in this context due to the possibility of reactivation in the setting of immunosuppression [9,10].

The relevance of this work can be perceived through the analysis of the vaccination rates before the IDC. By the time of the IDC, less than 5% of patients had all their vaccines up-to-date, according to their personal and health context. In the IDC, more than 650 vaccines were prescribed. These numbers illustrate the effort and commitment of ID physicians to increase vaccine uptake.

As expected, the most commonly prescribed vaccines in this adult population were those for the prevention of respiratory diseases (PCV13, PPV23, and influenza), followed by hepatitis A and B vaccines [11]. Other vaccines included MMR, tetanus, diphtheria, and pertussis; zoster (live); and chickenpox. After all, we consider that the IDC was (and can be in the future) an opportunity to review and update vaccines, including those not considered up-to-date by the National Vaccination Plan.

### 4.2. Vaccine Adherence

To the best of our knowledge, studies on the prescription and adherence rates to various vaccines among patients with IMIDs, including those with cancer, have not been reported so far. The available studies include data about vaccine uptake but not about its prescription [12,13]. In this sense, it has not been possible to estimate adherence since non-administered vaccines may be related to a lack of prescription or to a lack of adherence. Our study sheds light on this subject, as we collected data about adherence, focusing on vaccines that were prescribed but not administered.

Complete adherence to each of the vaccines for the prevention of respiratory diseases ranged between 69.1% (PPV23) and 87.3% (PCV13). Considering that, at baseline, the uptake of the PCV13 vaccine was less than 20%, these numbers are expressive and illustrate the impact of the IDC. A similar tendency was reported in a study that evaluated pneumococcal vaccination rates among IBD patients. In that case, the researchers registered an IDC-increased uptake from 16% at baseline to 86% [14]. Regarding hepatitis vaccines, adherence was similar for hepatitis A and B. The observed discrepancy between overall adherence and adherence to specific vaccines is explained by the number of vaccines prescribed for each patient (a mean of three vaccines).

### 4.3. Approaching Vaccine Hesitancy

Despite the efforts dedicated to prescribing vaccines in the IDC context and providing multidisciplinary follow-up, some patients did not comply with the prescriptions. In addition, although adherence to a specific vaccine was high, it was still far from optimal.

In this context, health care professionals dealing with vaccination must be sensitive when addressing patients’ knowledge, perceptions, and attitudes towards vaccines. Aspects like previous negative experiences, fear of adverse effects, fear of vaccine-induced flares of the baseline disease, and mistrust in scientific research and/or political authorities shall be considered and brought up for discussion with patients [12,13,15].

This study was conducted prior to the availability of the COVID-19 vaccine. The COVID-19 vaccination campaign has brought attention to the importance of vaccines and sparked discussions about vaccine safety, efficacy, and hesitancy [16]. Engaging in a broader discussion about the significance of vaccinations for global health may help alleviate COVID-19 vaccine hesitancy.

In this study, patients that had complete adherence were compared with those that did not to identify potential explanatory factors for vaccine hesitancy. No significant differences were found, but only sociodemographic and health-related determinants were assessed. This may suggest that patients’ knowledge, perceptions, and attitudes towards vaccines may play a major role in vaccine hesitancy and adherence, along with healthcare policies and socioeconomic barriers to access, including financial constraints. During the study period, the new recombinant zoster vaccine was not available in Portugal, and financial constraints precluded the prescription of the zoster (live) vaccine for those aged 50 years or older. This vaccine is the most expensive vaccine available in Portugal and is not supported by the Portuguese National Health System.

### 4.4. Improving Vaccine Adherence

In this study, ID physicians were responsible for infection screening and prevention, including vaccine prescription. Vaccination was performed according to the four standards for adult immunization practice of the Centers for Disease Control and Prevention (CDC): (1) assessment of vaccination status; (2) strong recommendation of needed vaccines; (3) offer or referral to a provider who can administer vaccines; and (4) delivery of the documentation related to administered vaccines [17]. The vaccine prescription was also in line with recommendations from European and American guidelines [2,3,4].

It has been often demonstrated that physician recommendation is the single most important factor in increasing vaccination rates [18]. Recent data showed that shared decision-making with patients increases the uptake of vaccines like influenza and pneumococcal [19]. In a recent systematic review, specific interventions also proved to increase vaccine uptake and preventive care adherence in IBD patients. These interventions were targeted at both patients and gastroenterologists. Regarding patients, the interventions included educational sessions, self-assessment questionnaires, and easier access to vaccines. Gastroenterologists were asked to participate in the vaccination process and were provided with tools to improve knowledge, recognize vaccine performance, and act as reminders [12]. A systematic review examined interventions to increase vaccine uptake in patients with rheumatoid arthritis [13]. These interventions focused almost exclusively on improving the prescription of vaccines and consisted mainly of reminder-type tools. The interventions proved to be effective in improving influenza, pneumococcal, and zoster vaccination rates.

It has already been discussed that consultations led by ID physicians may be a key intervention to increase vaccine prescription and adherence in patients with IMID [20]. In fact, ID physicians are one of the most competent groups of health care professionals to conduct these tasks, as they are trained in and have experience with infection prevention, diagnosis, and treatment strategies in immunocompromised patients.

Considering all this, the IDC shall occur as soon as possible after the IMID diagnosis and before the beginning of the immunomodulatory therapy.

Our findings suggest that achieving high or acceptable rates of vaccination in patients with IMID is possible, although it may be a challenging task. Infectious disease physicians may be able to take on this responsibility.

### 4.5. Limitations and Strengths

This study has limitations that must be acknowledged. First, the sample size is relatively small and consists mostly of patients with IBD and rheumatological diseases. Second, some data on sociodemographic factors are missing due to an incomplete EMR. And third, patients’ knowledge, perceptions, and attitudes towards vaccines, healthcare policies, and socioeconomic barriers to access were not assessed. However, this study presents several strengths that should be highlighted. To the best of our knowledge, this is the first prospective cohort study evaluating vaccine prescription and adherence in adult patients with IMID. In addition, vaccine prescription was undertaken by an ID physician in an IDC with the guarantee of appropriate and correct patient assessment in the context of pre-immunosuppression therapy. Moreover, the results of the study showed a high rate of prescription and satisfactory adherence to pneumococcal, influenza, and hepatitis vaccines. However, overall complete adherence was low, and more data on interventions to improve adherence to several different vaccines is needed.

## 5. Conclusions

In this study, vaccine prescription and adherence in patients with IMIDs were assessed using a real-world multidisciplinary clinical project between gastroenterologists, rheumatologists, and ID physicians. In the context of this multidisciplinary team, ID physicians were responsible for the screening and prevention of infection, including vaccination issues. The majority of the patients had complete adherence to the prescribed vaccines, and no significant differences were found in adherence related to sociodemographic and health-related determinants.

In conclusion, consultations led by ID physicians in the context of a multidisciplinary team may be a key intervention to increase vaccine prescription and adherence. We postulate that future studies shall focus on the perspective of patients towards vaccines, including beliefs, fears, knowledge, and financial constraints. This, along with specific and adequate interventions directed at patients and all healthcare providers, will provide information to implement tailored strategies to further improve adherence to vaccines.

## Figures and Tables

**Table 1 vaccines-11-00703-t001:** Baseline characteristics of patients at presentation to the IDC.

Sociodemographic Data	*n* (%)
Age, years (mean ± SD)	41 ± 15.1
Male gender	136 (51.9)
Education level (*n* = 156)	
Basic school degree	54 (34.6)
Secondary school degree	43 (27.6)
Higher education	59 (37.8)
Clinical data	*n* (%)
IMID	
Gastroenterology	215 (82.0)
Crohn’s disease	163 (62.2)
Ulcerative colitis	52 (19.8)
Rheumatology	47 (18.0)
Rheumatoid arthritis	21 (8)
Ankylosing spondylitis	9 (3.4)
Axial spondyloarthritis	7 (2.7)
Psoriatic arthritis	7
Polyarthritis	2 (0.8)
Juvenile idiopathic arthritis	1 (0.4)
Time from diagnosis to IDC, years (median (IQR))	3 (1–11)
Immunomodulatory therapy before IDC	
No therapy	95 (36.3)
On one drug	110 (42)
One two drugs	46 (17.6)
On three drugs	11 (4.2)
Class immunomodulatory drugs	
IS therapy	126 (48.1)
Biologic therapy	23 (8.8)
IS + biologic therapy	18 (6.9)
Time from therapy to IDC, months (median (IQR))	9 (1–75)

IDC: infectious diseases consultation; IQR: interquartile range; IMID: immune-mediated inflammatory disease; IS: immunosuppressive; SD: standard deviation.

**Table 2 vaccines-11-00703-t002:** Prescription and adherence to vaccines in 250 patients evaluated at the IDC.

Vaccine Prescription	*n* (%)
PCV13	212 (32.1)
Influenza inactivated	125 (18.9)
PPV23	110 (16.6)
Hepatitis A	98 (14.8)
Hepatitis B	91 (13.8)
Boosters	37 (5.6)
Full series	54 (7.4)
Measles, mumps, and rubella	8 (1.2)
Tetanus, diphtheria and pertussis	4 (0.6)
Zoster (live)	4
Chickenpox	3 (0.5)
Poliomyelitis inactivated	2 (0.3)
Yellow fever	1 (0.2)
Other	3
Total	661 (100)
Vaccine prescription by immunomodulatory therapy group	*n* (%)
IS therapy	118/126 (93.7)
Biologic therapy	22/23 (95.7)
IS + biologic therapy	17/18 (94.4)
No therapy	93/95 (97.9)
Vaccine adherence	*n* (%)
Complete (all vaccines)	151 (60.4)
Partial (at least one but not all)	
Two thirds or more	39 (15.6)
Between one to two thirds	31 (12.4)
One third or less	9 (3.6)
Null (none vaccine)	20 (8)
Total	250 (100)
Vaccine adherence by specific vaccine	*n* (%)
PCV13	185/212 (87.3)
Hepatitis B *	73/91 (80.2)
Hepatitis A	77/98 (78.6)
Influenza inactivated	96/125 (76.8)
PPV23	76/110 (69.1)

* Hepatitis B vaccine adherence was considered only if all prescribed doses were taken. IDC: infectious disease consultation; IS: immunosuppressive; PCV13: pneumococcal protein conjugate 13-valent vaccine; PPV23: pneumococcal polysaccharide 23-valent vaccine.

**Table 3 vaccines-11-00703-t003:** Comparison of vaccine adherence by specific vaccine according to immunomodulatory therapy.

	IS (*n* = 127)	Biologic (*n* = 23)	IS + Biologic (*n* = 18)	No Therapy (*n* = 94)	*p*-Value
PCV13, *n* (%)	92/99 (92.9)	14/19 (73.7)	9/15 (60)	70/79 (88.6)	0.001
Influenza, *n* (%)	47/63 (74.6)	10/11 (90.9)	4/8 (50)	35/43 (81.4)	0.162
PPV23, *n* (%)	35/53 (66)	5/6 (83.3)	8/9 (88.9)	28/42 (66.7)	0.463
Hepatitis A, *n* (%)	40/50 (80)	3/4 (75)	5/8 (62.5)	29/36 (80.6)	0.705
Hepatitis B *, *n* (%)	43/52 (82.7)	6/6 (100)	2/4 (50)	22/29 (75.9)	0.228
Live vaccines, *n* (%)	1/2 (50) **	0/0	0/0	8/12 (75) ***	1.000

* Hepatitis B vaccine adherence was considered only if all prescribed immunizations were taken. ** One patient was on prednisolone 5 mg daily, and the other was suspended from prednisolone to be vaccinated. *** Two patients were prescribed two live vaccines (adherence was only documented in one). IS: immunosuppressive; PCV13: pneumococcal protein conjugate 13-valent vaccine; PPV23: pneumococcal polysaccharide 23-valent vaccine.

**Table 4 vaccines-11-00703-t004:** Comparison of patients’ baseline characteristics according to vaccine adherence.

	Complete Adherence (*n* = 151)	Partial or No Adherence (*n* = 99)	*p*-Value
Age, years (mean ± SD)	41.7 ± 16.2	39.6 ± 13.3	0.265
Male gender, *n* (%)	77 (51)	52 (52.5)	0.813
Education level, *n* (%)			
Basic school degree	29 (32.6)	20 (33.9)	
Secondary school degree	24 (27)	18 (30.5)	
Higher education	36 (40.4)	21 (35.6)	0.822
IMID group, *n* (%)			
Gastroenterology	125 (82.8)	79 (79.8)	
Rheumatology	26 (17.2)	20 (20.2)	0.552
Time from diagnosis to IDC, years (median (IQR))	3 (1–10)	3.5 (1–13)	0.369
Immunomodulatory therapy group, *n* (%)			
IS therapy	78 (51.7)	40 (40.4)	
Biologic therapy	14 (9.3)	8 (8.1)	
IS + biologic therapy	8 (5.3)	9 (9.1)	
No therapy	51 (33.8)	42 (42.4)	0.244
Time from therapy to IDC, months (median (IQR))	9 (1–84)	10.5 (1–75)	0.894

IDC: infectious diseases consultation; IQR: interquartile range; IMID: immune-mediated inflammatory disease; IS: immunosuppressive; SD: standard deviation.

## Data Availability

The authors declare that all data supporting the findings of this study are available within the article and its Appendix A.

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
