# Peer review of "Adherence to Vaccines in Adult Patients with Immune-Mediated Inflammatory Diseases: A Two-Year Prospective Portuguese Cohort Study"

_vaccines, 2023, doi:10.3390/vaccines11030703_

Round 1

Reviewer 1 Report

This study is quite interesting, although based on a modest sample.

The study is well described, I have made comments directly in the PDF to improve the understanding by the reader.

It is unfortunate that the authors performed a relatively simple and descriptive statistical analysis with their data. A more sophisticated statistical analysis would surely have yielded interesting results. 

Author Response

Authors’ response to reviewer 1

Title: Adherence to vaccines in adult patients with immune-mediated inflammatory diseases: a two-year prospective cohort study (Vaccines- 2245941)

Corresponding Author: Cândida Abreu

We express our gratitude to the reviewer for their insightful suggestions, which we have taken into careful consideration. As a result, we have implemented the necessary corrections and clarifications in the revised version of the manuscript.

  1. Title : the place were the study took place

Now in title:

Adherence to vaccines in adult patients with immune-mediated  inflammatory diseases: a two-year prospective Portuguese cohort study

  1. Abstract: add a sentence about how these patient’s were recruited

Now in Abstract:

This prospective  cohort study included 262 consecutive adults with inflammatory bowel disease and rheumatological diseases who underwent an infectious diseases evaluation before initiating or switching immunosuppressive/biological therapy.

  1. Introduction

..Introduction could be more developed so that context and interest of the study is more highlighted

We express our gratitude to the Reviewer for bringing this observation.

Now in the Introduction:

Our manuscript aims to share our experience with this cohort and ultimately increase vaccination rates. We believe that a multidisciplinary approach directed towards patients with IMIDs is a robust strategy for surveillance and treatment. Infection screening and prevention are naturally the primary focus of ID physicians, making them well-suited for vaccine-related tasks. However, we went beyond merely prescribing vaccines and emphasized adherence to vaccine prescriptions in adult patients with IMIDs assessed in the context of a consultation with an infectious disease (ID) physician.

3.. Methods

3.1 –  There was no sample size calculated? It is a pragmatic sample or only the inclusion criteria was the gateway to be included? if so it must be explained

We thank the Reviewer for the question.

The sample size was not calculated. We did not calculate the statistical power of the study because in the worst-case scenario, vaccine adherence is 60%. Using this value as a reference (as being the worst), the power of the study is 88%, which means that for vaccines with higher adherence, it is even higher.

Now in Methods:

We enrolled all eligible consecutive patients who met the inclusion criteria and provided their informed consent to participate in the study during the study period.

3.2 Ethics statement

Now in Methods:

The study was approved by the Ethics Committee of CHUSJ (356/2018). Informed written consent, in accordance with the Declaration of Helsinki, was  obtained after explaining the procedures to each participant. The participants  were able to leave the research at any time without any consequences, and the individuals who decided not to participate received the same treatment offered  to participants. The method of data collection allowed that demographic  information was integrated in the evaluation, while assuring confidentiality.

On behalf of all authors, sincerely

Cândida Abreu.

Reviewer 2 Report

Dear editor,

Thank you for the kind invitation to review this manuscript. It is generally well written and easy to follow. My comments are appended below

The authors describe a study which evaluated adult IMID patients as well as their adherence to vaccines. 

Introduction 

- What is suboptimal vaccination rates for IMID patients?

- What are the literature performed regarding vaccination for IMID patients and the specific knowledge gap in literature can be better highlighted to enhance the importance of this study.

Methods

- Suggest to use a checklist for cohort study e.g. JBI or CASP

- What is the statistical power of the study

- How are vaccination records for IMID patients kept?

-> Is there a central immunization registry for patients? 

- Are serology testing done to assess if vaccination is required for vaccination such as hepatitis A and B?

- Are there routine childhood immunisation programs in the country?

-> It would be helpful to put in some small details as well as any specific immunisation programs for IMID patients 

Results

- Suggest to combine table 1 and 4 to minimize need for duplicated content

Discussion

- How does the study compare to other studies performed in patients with rheumatological conditions / cancer patients

- What would be the most important implication of the study?

- The importance of covid-19 vaccination cannot be undermined 

-> This is important given the high covid-19 vaccination hesitancy. 

-->https://pubmed.ncbi.nlm.nih.gov/34452026/

Author Response

Authors’ response to reviewer 2

Title: Adherence to vaccines in adult patients with immune-mediated inflammatory diseases: a two-year prospective cohort study (Vaccines- 2245941)

Corresponding Author: Cândida Abreu

We would like to thank the Reviewer for his comments and suggestions, which were thoroughly accepted. We have introduced the corresponding corrections and clarifications on the new version of the manuscript.

  1. Introduction 
    • - What is suboptimal vaccination rates for IMID patients?

Suboptimal vaccination was used in the sense  that a high percentage of IMID patients are not vaccinated against vaccine preventable diseases as desirable (references 6,7) and according to published vaccine recommendations (references 2-5).

Now in text

Even though, adult patients with IBD and rheumatological diseases still have suboptimal vaccination rates.

  • - What are the literature performed regarding vaccination for IMID patients and the specific knowledge gap in literature can be better highlighted to enhance the importance of this study.

We would like to express our gratitude to the Reviewer for the question. Our manuscript aims to share our experience with this cohort and ultimately increase vaccination rates. We believe that a multidisciplinary approach directed towards patients with IMIDs is a robust strategy for surveillance and treatment. Infection screening and prevention are naturally the primary focus of ID physicians, making them well-suited for vaccine-related tasks. However, we went beyond merely prescribing vaccines and emphasized adherence to vaccine prescriptions.

Now in the introduction:

This prospective cohort study aimed to go beyond vaccine prescription and evaluate the adherence to vaccines in adult patients with IMIDs assessed in the context of a consultation with an infectious disease (ID) physician.

  1. Methods

2.1 - Suggest to use a checklist for cohort study e.g. JBI or CASP

Now in Methods:

Quality appraisal using critical appraisal skill program (CASP) checklist for cohort studies was included (supplementary Table 1).

Supplementary Table 1 - Quality appraisal using CASP checklist for cohort studies

CASP cohort study checklist

Answer

Did the study address a clearly focused issue?

Yes

Did the study address a clearly focused issue?

Yes

Was the cohort recruited in an acceptable way?

Yes

Was the exposure accurately measured to minimise bias? 

Yes

Was the outcome accurately measured to minimise bias?

Yes

Have the authors identified all important confounding factors?

Yes

Have they taken account of the confounding factors in the design and/or analysis?

Yes

Was the follow up of subjects complete enough?

Yes

How precise are the results?

Can’t tell

Do you believe the results?

Yes

Can the results be applied to the local population?

Yes

Do the results of this study fit with other available evidence?

Can’t tell

Does the study have implications for practice?

Yes

2.2 - What is the statistical power of the study

We did not calculate the statistical power of the study because in the worst-case scenario, vaccine adherence is 60%. Using this value as a reference (as being the worst), the power of the study is 88%, which means that for vaccines with higher adherence, it is even higher.

Now in Methods:

We enrolled all eligible consecutive patients who met the inclusion criteria and provided their informed consent to participate in the study during the study period.

2.3 - How are vaccination records for IMID patients kept?  Is there a central immunization registry for patients? 

Now in the Methods:

The vaccines that were administered have been recorded in the national vaccine registry platform of the Portuguese National Health Care System. In rare cases, some vaccines may have been recorded in an individual vaccination book from the same system.

2.4 - Are serology testing done to assess if vaccination is required for vaccination such as hepatitis A and B?

Always; the answer is under the title vaccines prescription (page 2 of 9)

2.5 - Are there routine childhood immunisation programs in the country?

Since 1965, there has been a National Vaccination Program, now includes 13 different vaccines, (Programa Nacional de Vacinação - eVacinas).

2.6 -  It would be helpful to put in some small details as well as any specific immunisation programs for IMID patients 

 On the text: Vaccine prescriptions were carried out in accordance with the recommendations of EULAR, AGA, and ECCO (references 2-4), as following: hepatitis A vaccine….

  1. Results

3.1 - Suggest to combine table 1 and 4 to minimize need for duplicated content

 The authors believe that it would be preferable to keep the two tables separate, as they have different focuses: Table 1 covers demographic and clinical aspects, while Table 4 focus on vaccine adherence.

  1. Discussion

4.1 - How does the study compare to other studies performed in patients with rheumatological conditions / cancer patients

In the text: To the best of our knowledge, studies on the prescription and adherence rates to various vaccines among patients with IMIDs, including those with cancer.

4.2 - What would be the most important implication of the study?

In the text: Our findings suggest that achieving high or acceptable rates of vaccination in patients with IMID is possible, although it may be a challenging task. Infectious disease physicians may be able to take on this responsibility.

4.3 - The importance of covid-19 vaccination cannot be undermined; this is important given the high covid-19 vaccination hesitancy. (>https://pubmed.ncbi.nlm.nih.gov/34452026/)

In the text: This study was conducted prior to the availability of the COVID-19 vaccine. The COVID-19 vaccination campaign has brought attention to the importance of vaccines and has sparked discussions about vaccine safety,  efficacy and hesitancy16. Engaging in a broader discussion about the significance of vaccinations for global health may help alleviate Covid-19 vaccine hesitancy.

  1. Aw J, Seng JJB, Seah SSY, Low LL. COVID-19 Vaccine Hesitancy-A Scoping Review of Literature in High-Income Countries. Vaccines (Basel). 2021 Aug 13;9(8):900. doi: 10.3390/vaccines9080900

On behalf of all authors, sincerely

Cândida Abreu.

Round 2

Reviewer 1 Report

thanks to the authors for their corrections and additions. This article is acceptable in this form

Reviewer 2 Report

Dear authors and editor,

The changes are satisfactory. Nil further comments from me